# Numerical Study on the Effect of Z-Warps on the Ballistic Responses of Para-Aramid 3D Angle-Interlock Fabrics

**DOI:** 10.3390/ma14030479

**Published:** 2021-01-20

**Authors:** Yingxue Yang, Xiuqin Zhang, Xiaogang Chen, Shengnan Min

**Affiliations:** 1Beijing Key Laboratory of Clothing Materials R & D and Assessment, Beijing Engineering Research Center of Textile Nanofiber, School of Materials Design & Engineering, Beijing Institute of Fashion Technology, Beijing 100029, China; yyx5928@163.com (Y.Y.); clyzxq@bift.edu.cn (X.Z.); 2Department of Materials, University of Manchester, Manchester M13 9PL, UK; xiaogang.chen@manchester.ac.uk

**Keywords:** para-aramid fabrics, Z-warps, ballistic responses, finite element method

## Abstract

In order to achieve an efficient ballistic protection at a low weight, it is necessary to deeply explore the energy absorption mechanisms of ballistic fabric structures. In this paper, finite element (FE) yarn-level models of the designed three-dimensional (3D) angle-interlock (AI) woven fabrics and the laminated two-dimensional (2D) plain fabrics are established. The ballistic impact responses of fabric panels with and without the interlocking Z-warp yarns during the projectile penetration are evaluated in terms of their energy absorption, deformation, and stress distribution. The Z-warps in the 3D fabrics bind different layers of wefts together and provide the panel with structural support along through-the-thickness direction. The results show that the specific energy absorption (SEA) of 3D fabrics is up to 88.1% higher than that of the 2D fabrics. The 3D fabrics has a wider range of in-plane stress dispersion, which demonstrates its structural advantages in dispersing impact stress and getting more secondary yarns involved in energy absorption. However, there is a serious local stress concentration in 2D plain woven fabrics near the impact location. The absence of Z-warps between the layers of 2D laminated fabrics leads to a premature layer by layer failure. The findings are indicative for the future design of ballistic amors.

## 1. Introduction

Materials such as para-aramid, ultra-high-molecular-weight-polyethylene (UHMWPE), carbon and glass fibers possess high strength and modulus, as well as excellent chemical stability, which can well meet the requirements of light-weight and high performance for ballistic applications [1,2]. These linear fibers are often woven into planar fabrics for further employment as soft armors or reinforcement for composites. The weave structure of high-performance fabrics is one of the factors that affect their ballistic performance. Reasonable structural design can not only improve the ballistic performance, but also reduce the weight of ballistic resistant products.

The structures of three-dimensional (3D) angle-interlock (AI) fabric, through-the-thickness angle-interlock (TTAI) and laminated two-dimensional (2D) plain weave fabrics are shown in Figure 1. Lacking of binding between different weft layers, the 2D fabrics need to be layered up to achieve an equivalent thickness and areal density to the 3D structures. Comparisons on the ballistic performance of different 2D weaves were reported before [3,4]. The conventional 2D plain and basket weave fabrics are orthotropic structures, which are mechanically balanced along both warp and weft directions [5]. However, the laminated 2D fabric reinforced composites are prone to excessive delamination, resulting in large indentations [6]. Yang and Chen [7] found that the plain fabric layers at different positions in the laminated panel exhibit different transverse deformation and stress distribution. Understanding the failure modes of different fabric layers will help to improve the ballistic performance. The Z-warp (interlocking warp) yarns in 3D fabrics bind the weft layers together, providing enhancement through the panel thickness, and effectively suppressing large local deformations upon impact [8]. Moreover, previous studies shown that the 3D woven through-the-thickness angle-interlock fabrics exhibit excellent mouldability, which can reinforce complex doubly-curved shapes, such as female body armor and ballistic helmets, in single-piece [9,10,11]. However, the differences in the ballistic responses and failure mechanisms of the 2D plain and 3D angle-interlock fabrics are still not understood.

The penetration of projectile into fabrics is a complex nonlinear transient process, which is difficult to examine by experimental methods. The ballistic responses and impact energy absorption mechanisms of the fabrics can be further studied with the help of finite element (FE) modeling [12,13,14,15,16,17,18,19]. In the cases of fabric modeling, the macro-scale models often neglect the yarn geometry by homogenize the fabric structure, and the micro-scale models include too much details by modeling the filaments reactions within the yarns. Compared to these two types of models, the meso-scale models reflect the real yarn geometry at a reasonable computing time. During the process of projectile penetration, the yarns within a fabric will be deformed, fractured, slipped, and pulled-out to dissipate and absorb the impact energy [20]. The meso-scale yarn level FE models can help to analyze the stress propagation mechanisms in yarns more accurately. Chu et al. [14] and Zhou et al. [15] found in their simulations that the friction between yarn and yarn, yarn and projectile in plain woven fabrics is an important form of energy absorption. The energy absorption of the fabrics can be increased by controlling the inter-yarn friction in a certain range. Wang et al. [16] studied the influences of crimp in plain woven fabrics. The results showed that the yarn crimp is a key factor affecting stress wave propagation, and fabrics with small yarn crimp is beneficial to the stress wave propagation. Hou et al. [17] also found that the yarn crimp coefficient and weave density are the key factors influencing 3D fabric ballistic behavior. Jia et al. [18] found that the structural stability of 3D orthogonal fabric depends on the Z-warp yarns, although the Z-warp yarns absorb much less energy than weft yarns. The role of the Z-warps in dissipating the impact stress were not examined. Kedzierski et al. [19] studied the blunt trauma resistance of UD laminates, 2D plain fabrics and 3D multiaxial fabrics under low-velocity impacts by experimental and numerical methods. According to the normalized results, the ballistic protection of 3D multiaxial fabrics is the best, whereas that of 2D plain fabrics being the worst. Whereas the performance of these fabrics under ballistic impacts are to be investigated.

Having the potentials in reinforcing protective equipment with complex shapes and the enhanced through-the-thickness properties, it is useful to evaluate the ballistic performance of 3D angle-interlock fabrics in comparison with the commonly used 2D plain weave fabrics. The ballistic impact behavior of 2D plain fabrics has been studied intensively by various researchers, while the failure mechanisms of 3D fabrics during ballistic penetration has not been systematically studied in comparison, especially the role of Z-warps in the 3D fabrics. The yarn-level FE models were established in this study to analyze the structural responses of the yarns in 3D angle-interlock fabric and 2D plain weave fabric laminated panels during projectile penetration. The results will provide theoretical guidance for the design of ballistic protective fabric and composite panels.

## 2. Finite Element Models for Fabrics under Ballistic Impact

In order to simulate the transient process of the projectile penetration to fabrics and clarify the ballistic responses of different yarns during penetration, four FE models at yarn-level are established, as shown in Figure 2. The models are 3-ply and 5-ply laminated plain fabrics (3-plain, 5-plain), 5-weft-layer AI fabric (5-AI), and 4-weft-layer TTAI fabric (4-TTAI). The weave specifications of the fabrics are shown in Table 1. In all cases, the projectile hits the center of the fabric. Due to the symmetrical nature of plain weave fabrics, only 1/4 of the fabric and projectile geometry model was established for 3-plain and 5-plain to save the calculation time. The model of 4-TTAI is built to verify the accuracy of designed FE model. The areal density and thickness of 3-plain and 5-AI are close, and are used to compare the energy absorption efficiency of the two structures. The 5-plain and 5-AI have the same number of weft layers, so the analyses based on the number of weft layers will be more intuitive.

The fabric models are assembled by warp and weft yarns. The cross-section of a single yarn is a lenticular shape composed of two symmetrical arcs [21]. The geometric parameters of the yarns and the geometric relationships of the fabrics are obtained by measuring the fabrics under magnifying microscope and are listed in Figure 3. The diameter and height of the cylindrical projectile are both 5.5 mm and the mass of the projectile is 1.0 g.

The material properties are shown in Table 2, according to the results in [22]. The yarns are assumed to be homogeneous and isotropic although the real yarns are made of continuous filaments and orthotropic [23]. Previous studies and ours [24,25,26] have shown that the homogeneity and isotropy assumptions of the yarn lead to acceptable results with 2.4% difference in energy absorption. Despite some differences in mechanical behavior of the material between quasi-static loading and dynamic loading, many have used the data under quasi-static loading in simulations for simplicity.

The yarn and projectile are meshed with eight-node solid brick elements, as shown in Figure 4. Mesh sensitivity studies for various element sizes have suggested that using ten solid elements across the yarn section is sufficient for this analysis. Since the stiffness of the projectile is much higher than that of the yarn, it can be regarded as a rigid body. In the model, a global friction coefficient is set to be 0.15 [22]. All of the fabric structures are compared under the same value of friction coefficient. The effect of yarn–yarn friction on energy absorption can be considered to be at an equivalent level. The projectile penetrates from the center of the fabric at the impact velocity of 480 m/s, as it is the average velocity that can be achieved by the ballistic range (with impact velocity of 445–520 m/s) used when conducting the experimental studies. The fabric samples are clamped at their edges and a symmetrical boundary conditions is applied to the cross-sections in 1/4 plain model.

The accuracy of FE models can be verified according to the ballistic experimental results [27]. The experimental ballistic tests evaluate the impact and residual velocities of the projectile before and after penetrating the fabric panels. Therefore, the residual velocities under different impact velocities were compared for validating the FE model. An FE model of 4-TTAI fabric with thread densities of 12 ends/cm and 30 picks/cm was established and the penetration processes of projectile at a relatively high velocity range were simulated. The energy dissipation due to projectile deformation, fiber intermolecular friction, air resistance and acoustic losses are all assumed to be negligible. In comparison to the experimental and the numerical data, the model has shown good agreement in matching the values of projectile residual velocity, with the linear regression curve fitted as shown in Figure 5. The fabric model for the plain weave fabric was established and validated in the previous research [22]. The validated fabric models are used to simulate and analyze the ballistic responses.

## 3. Results and Discussion

During the impact process, the fabric will absorb and dissipate the impact kinetic energy carried by the impact projectile in the form of vibration, deformation or breakage, and friction. The ballistic performance information obtained through FE analyses are listed in Table 3.

### 3.1. Time History of the Projectile

The model analysis starts at the projectile input at 480 m/s. The projectile decelerates as it perforating the fabric panel and the residual velocity is defined as the projectile velocity with changes less than 0.05 m/s within 3 µs. The moment at which the projectile reaches its residual velocity is defined as the panel failure time where the projectile penetration is completed. The projectile residual velocity of 3-plain is 4.3% higher than that of 5-AI at a 42.7% larger weight. The 8.4 μs earlier failure time of 3-plain, indicating worse impact resistance and smaller absorption capacity when compared to the 3D 5-AI. The projectile peak acceleration of 5-AI is 6.9% higher than that of 3-plain, which means the 5-AI provided larger reaction force in resisting the projectile penetration.

### 3.2. Energy Absorption

The impact energy absorbed (*E_a_*) by the fabric can be considered as the kinetic energy lost by the projectile, which can be calculated according to the initial velocity (*v_i_*) and residual velocity (*v_r_*) before and after the projectile penetrates the fabric. *E_a_* is formulated as follows:(1)$$E_{a}\;=\;\frac{1}{2}m\;(v_{i}^{2}-v_{r}^{2})$$
where *m* is the mass of the projectile.

Specific energy absorption (*SEA*) refers to the projectile impact energy absorbed by the fabric per unit areal density, which is an important parameter for evaluating the energy absorption efficiency of the fabric. The calculation of SEA follows the formula below:(2)$$\mathrm{SEA}\;=\;\frac{E_{a}}{A}$$
where *A* is the areal density of the fabric. 

As shown in Table 3, the *SEA* of 3D fabric 5-AI is 75.5–88.1% higher than that of 2D fabrics, indicating that 5-AI fabric have higher energy absorption efficiency than the multiply plain weave fabrics under the normalized areal density.

To further understand the energy absorption mechanisms, the contribution of projectile impact kinetic energy absorption is analyzed in forms of fabric kinetic energy (KE), total strain energy (IE), frictional dissipative energy (FD), and others as presented as Equation (3). As mentioned previously, the other sources of energy dissipation (including projectile deformation, fiber intermolecular friction, air resistance and acoustic losses) were assumed to be negligible. Among them, the fabric IE mainly includes its recoverable strain energy (SE), plastic deformation energy (PD), the inevitable artificial strain energy (AE) and other neglectable in the simulation process, as shown in Equation (4).
(3)Total Energy = KE + IE + FD + other
(4)IE = SE + PD + AE + other

As can be seen from Figure 6a, when impacted by the projectile, the fabric mainly converts the kinetic energy of projectile into IE and KE. The contribution of FD is relatively small (15.3–18.6% of the total energy absorption) and at an equivalent amount among these three fabrics. Generally, the IE accounts for a larger proportion in the laminated 2D plain fabric panels, which is 1.5–2.1 times of KE. Figure 6b depicts the contribution of different forms in IE. The SE is 38.6% higher than PD for the angle-interlock panel. While the PD contributes more in both 2D fabric panels. According to Chu et al. [28], the primary yarns that in direct contact with the projectile tends to fail by plastic fracture damage, while the secondary yarns interlaced with the primary yarns participate in energy absorption by tensile failure during the penetration process. The fact that SE contributes more in 3D fabric indicates that there are more secondary yarns involved in energy absorption of the fabric, which leads to a wider fabric deflection and an improved energy absorption capability.

The time history of strain energy absorbed by each form was extracted. The two 2D fabric panels present a similar trend, therefore, only 3-plain and 5-AI fabrics are presented in Figure 7. It can be seen that the PD of the two fabrics presents a trend of continuous growth, while the SE gradually decreases after reaching its peak. Compared with 3-plain, the SE of 5-AI is higher, which means that the yarns of 5-AI can store more energy in a manner of elastic deformation. The strained yarns would move against each other relatively, resulting in the conversion of SE to KE and FD for a gradual release. It would also induce yarn fracture after reaching the maximum strain value, which is related to the plastic deformation of materials and energy absorption by PD.

The 5-plain and 5-AI have the same number of weft layers. To further understand the yarn failure within different weft layers, the failure time, deformation and stress distribution of the two types of the fabric panels are compared in Section 3.3, Section 3.4 and Section 3.5.

### 3.3. Fabric Deformation

The failure time of the primary yarns in each layer of 5-AI and 5-plain is compared, as shown in Figure 8. Generally, the failure time of each layer of yarns in 5-plain is 3–6 μs earlier than that of 5-AI. A layer-by-layer fracture sequence is also noticed in the plain laminated panel. While for the 5-AI, the primary yarns in the rear layers break together, acting as a unit against the projectile. Based on the fact that the weft layers in 5-AI are bound together by the Z-warp yarns, it is speculated that Z-warps in 5-AI provide structural support for the panel through-the-thickness, so that a wider range of yarns can disperse the impact stress together and thus have better penetration resistance. The study of the impact stress distribution of the two fabrics will prove this speculation, as it will be shown in Section 3.5.

### 3.4. Deformation of Fabric Panels

The in-plane and transverse deformations are extracted from different viewports. The in-plane deformations are described by the range of depression along warp and weft directions. The transverse indentations are the depths of the back-face deformation.

The sizes of the panel in-plane deformation dispersion along the primary warps and wefts were extracted to describe the range of primary yarn involvement. It can be seen from Figure 9 that the deformation of 5-plain is equivalent along both warp and weft directions, while the weft deformation range of 5-AI is 81.8% larger than that of the warp direction. This is because the interlocking warps in 5-AI have a higher level of crimp than the weft yarns, which creates a rather straight channel for stress propagation along the weft yarns. Whereas the yarn crimp is at the same level in the plain fabrics. 

Figure 10 outlined the displacement of 5-AI and 5-plain panels at their failure moment. The size of the back-face depression of 5-plain is smaller than that of the 5-AI and its stress is more concentrated, which is consistent with the in-plane stress distribution results. Due to the lack of interaction between the layers of 5-plain, an obvious delamination between layers can be noticed and the depth of the back-face deformation is larger. While the weft layers in 5-AI are still connected after penetrated by the projectile with the existence of Z-warps, and the indentation is 7.7% smaller than that of the 5-plain. Therefore, the structure of 3D 5-AI fabric can more effectively suppress the back-face deformation when compared to the 2D plain fabric laminated structure, owing to the cohesion of the Z-warp yarns.

### 3.5. Stress Distribution

The impact induced strain energy is dissipated along the fibers and propagates in the forms of transverse wave and longitudinal wave. The longitudinal wave propagates along the fiber axis, while the transverse wave propagates along the thickness direction. The propagation velocity of longitudinal wave is mainly determined by the properties of fiber materials, while that of transverse wave is also related to the strain of fiber, which is usually smaller than the longitudinal wave [29]. To further understand the energy absorption advantages of 3D fabrics, the stress distribution on fabrics were extracted from the yarn-level model to deeply study the differences between 5-AI and 5-plain in terms of damage energy absorption mechanisms and dynamic stress responses.

The stress distribution on the back face of the quarter fabric panels was examined after the penetration of projectile. A quarter of the fabric panels with impacts located at the upper-left corner were examined in both warp and weft directions. Figure 11 provides the vertical views of stress distribution on 1/4 5-AI and 5-plain. In these models, the warp and weft yarns are in line with the X and Z axis, respectively. It can be seen clearly that the in-plane stress of 5-plain mainly concentrates on the impact point, and the range of stress distribution is small. However, 5-AI has a wider range of stress spread in weft direction, and there is an uneven stress distribution in warp and weft direction, due to the difference in the warp and weft density of 5-AI, as listed in Table 1. This agrees with the findings in fabric deformation in Section 3.4.

#### 3.5.1. In-Plane Stress Propagation

Based on the findings in Section 3.3, the yarns at the impact face failed early (see Figure 8), so the stress distribution of 5th weft at different moments was extracted to better analyze the in-plane longitudinal wave propagation in fabrics, as shown in Figure 12. It can be seen that the stress distribution of 5-plain is mainly concentrated near the impact point. As time goes on, the 5-plain will reach its peak stress earlier, leading a premature breakage of back-face yarns. The range of in-plane stress propagation in the back-face yarns of 5-AI is wider, which proves that more secondary yarns that interlaced with the primary yarns in the 3D fabric will participate in dispersing the impact stress.

#### 3.5.2. Stress Propagation through the Panel Thickness

To compare the velocity differences of transverse wave along the penetration direction between 2D and 3D fabric panels, the stress responses of different locations away from the impact point were extracted. Based on the stress distribution shown in Figure 11, the in-plane stress distribution in the 3D fabric is uneven, where the weft direction has a larger deformation range than the warp direction. For further comparison, 3 nodes were selected in each model, as shown in Figure 13a,b: A_3d_, B_3d_ and C_3d_ are selected from 5-AI along the weft direction, and A_2d_, B_2d_ and C_2d_ from 5-plain. It can be seen from Figure 13a,b that node A_3d_, A_2d_, B_3d_, and B_2d_ locate on the impact surface of the fabric panels with node A_3d_ and A_2d_ closer to the projectile. The distance of node A_3d_ and A_2d_, node B_3d_ and B_2d_ to the impact center are about 4.2 mm and 8.2 mm, respectively. Node C_3d_ and C_2d_ locates on the back-face of the panel 1.9 mm and 2.1 mm underneath node A_3d_ and A_2d_, respectively. Figure 13c,d shows the time history of stress response of the three nodes in 2D and 3D fabrics from the weft direction. It can be seen that in both 2D and 3D fabrics, point A_3d_ and A_2d_ respond the fastest when the projectile impacts. The stress wave, respectively, propagates to node B_3d_ and B_2d_ before arriving at node C_3d_ and C_2d_ along the thickness direction. This agrees with findings that the propagation velocity of longitudinal waves is higher than transverse waves [29]. It is also noticed that the stress of all three nodes in 5-AI are at a similar level throughout the penetration process. Whereas the stress experienced in 5-plain indicates a concentration on the impact face near the impact location, as the stress in node B_2d_ being the lowest. This is indicative to the design of structural hybrid panels. The use of 3D fabrics at the impact face will lead to a higher resistance and help to disperse the impact stress into wider ranges.

Furthermore, the stress responses of node C_3d_ and C_2d_ in 5-AI and 5-plain are compared in Figure 14. It can be seen that the responding time of node C_3d_ is about 1 µs later than C_2d_, meaning that the capacity of stress propagation through-the-thickness direction in 5-AI is relatively lower than that of the 5-plain, which is mainly because of the higher level of Z-warps crimp and the less interlacements between the layers in the 3D fabric. 

The findings of current studies provide guidance for future design of ballistic fabrics, as large crimps (interlocking depth) of the Z-warps would limit the efficiency of stress wave propagation through the panel thickness.

## 4. Conclusions

The differences of ballistic responses between 3D and 2D para-aramid fabrics are compared by finite element method to study the effect of the Z-warps. The conclusions are as follows:The overall ballistic protection performance of 3D AI fabric is better than that of laminated 2D plain fabrics by exhibiting an up to 88.1% higher SEA and a 7.7% smaller indentation.When impacted by a projectile, the 3D fabric is more inclined to convert the impact kinetic energy of the projectile into elastic strain energy, indicating more secondary yarns involvement in energy absorption of the 3D fabrics.While the 2D fabrics present earlier failure layer by layer, the 3D AI fabric has a smaller indentation and 8.4 μs longer resistance time than 5-plain, due to the cohesion of the Z-warp yarns. Therefore, the structure of 3D fabric can more effectively suppress the back-face deformation and possess better penetration resistance.3D AI fabric has a wider range of in-plane stress propagation in the panel back-face, while the laminated 2D fabrics experiences a stress concentration on the impact surface. It indicates that the 3D fabric has a better capacity of in-plane stress propagation when compared with 2D fabrics.Compared with the 2D fabric, the responding time of the back-face node in 5-AI is about 1 µs later due to the higher level of Z-warps crimp and the less interlacements between the weft layers in the 3D fabric.

The findings of current research are indicative for the design and construction of soft armor panels and the composites reinforced by fabrics.

## Figures and Tables

**Figure 1 materials-14-00479-f001:**
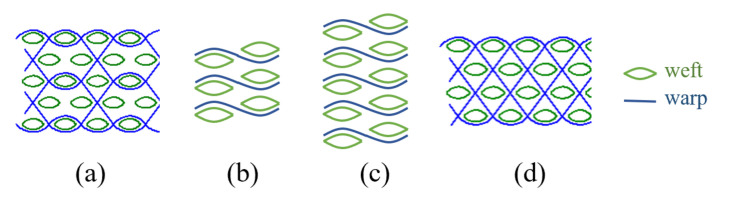
Side views of the fabric panel construction of (**a**) single-ply 3D AI fabric with 5 weft layers; (**b**) 3-ply laminated 2D plain weave fabrics; (**c**) 5-ply laminated 2D plain weave fabrics, and (**d**) single-ply 3D TTAI fabric with 5 weft layers.

**Figure 2 materials-14-00479-f002:**
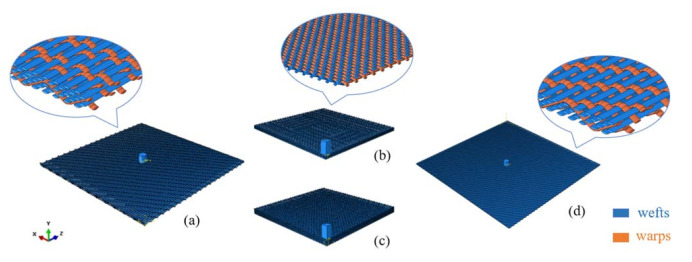
Finite element models: (**a**) 5-AI; (**b**) 1/4 geometry 3-plain; (**c**) 1/4 geometry 5-plain; (**d**) 4-TTAI.

**Figure 3 materials-14-00479-f003:**
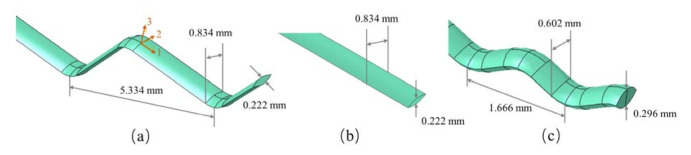
Geometrical parameters of yarns: (**a**) the Z-warp of 5-AI; (**b**) the weft yarn of 5-AI; (**c**) the warp and weft of plain weave fabrics.

**Figure 4 materials-14-00479-f004:**
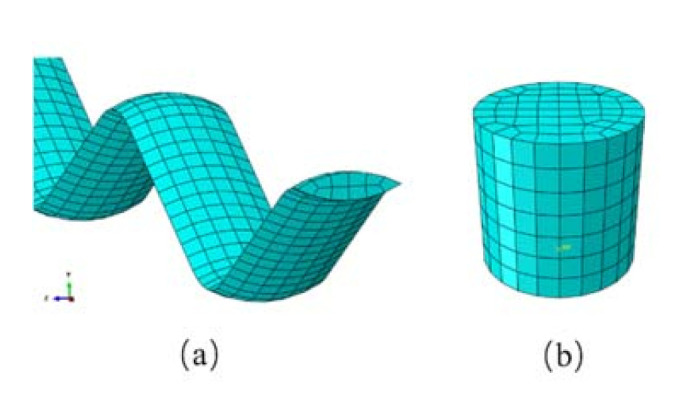
Mesh for the (**a**) yarn and (**b**) projectile.

**Figure 5 materials-14-00479-f005:**
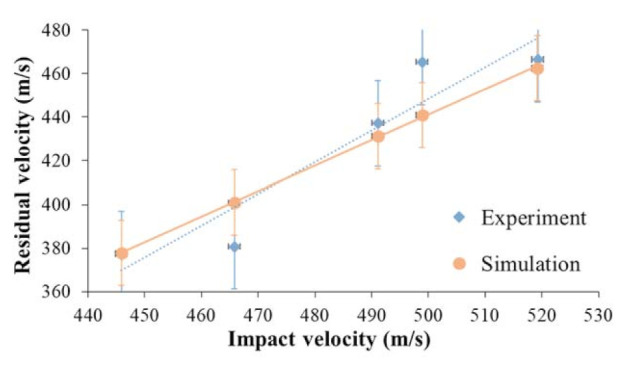
FE model validation against experimental results.

**Figure 6 materials-14-00479-f006:**
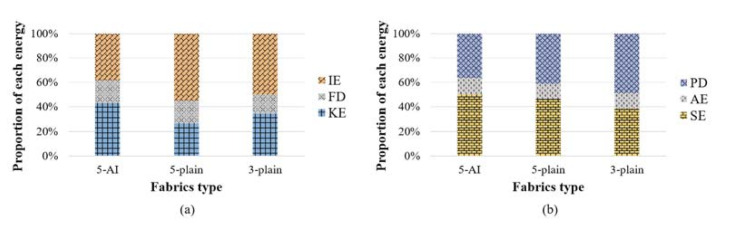
The contribution of each energy form in (**a**) the total energy and (**b**) the IE (total strain energy) of the fabric panels.

**Figure 7 materials-14-00479-f007:**
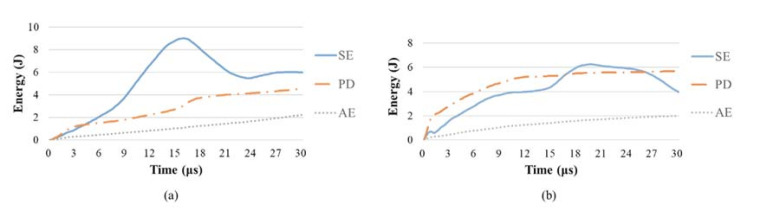
Time history of different forms of strain energy in (**a**) 5-AI and (**b**) 3-plain.

**Figure 8 materials-14-00479-f008:**
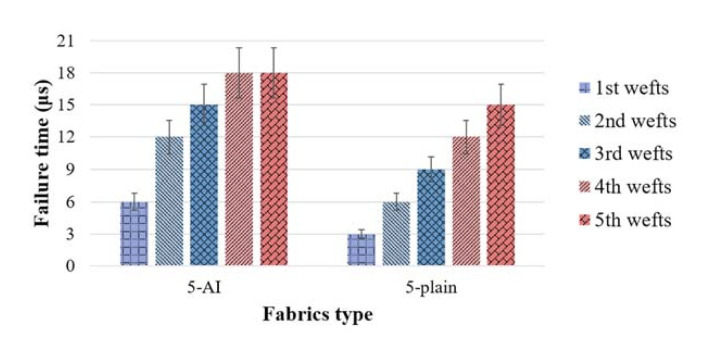
The failure time of each weft layer in the fabric panels.

**Figure 9 materials-14-00479-f009:**
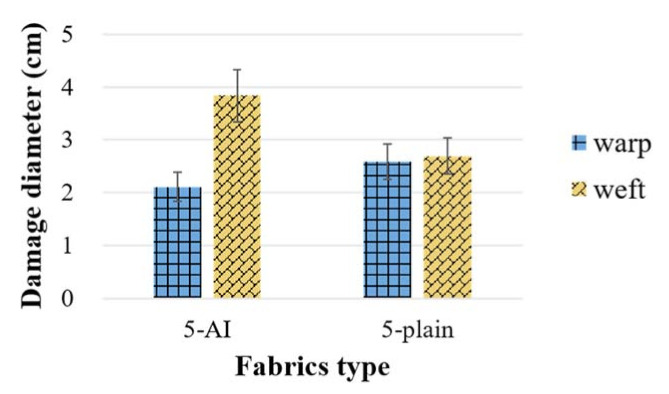
Damage size of fabric panels at the back face.

**Figure 10 materials-14-00479-f010:**
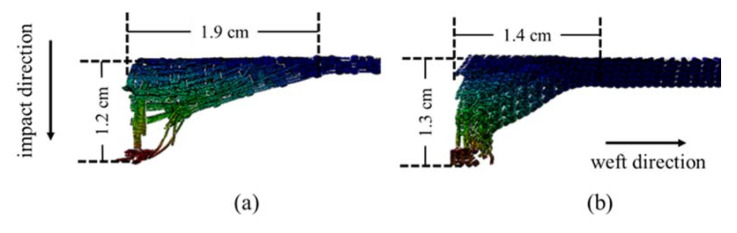
Side view of deformation along the weft direction in (**a**) 5-AI and (**b**) 5-plain.

**Figure 11 materials-14-00479-f011:**
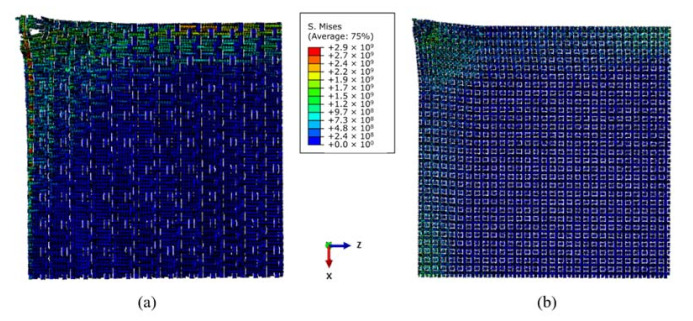
Range of back face stress distribution on 1/4 geometric model of (**a**) 5-AI and (**b**) 5-plain.

**Figure 12 materials-14-00479-f012:**
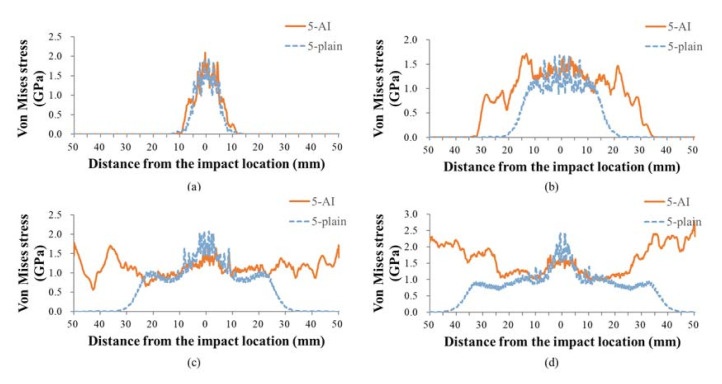
The stress distribution in 5th weft yarns at (**a**) 3 μs; (**b**) 6 μs; (**c**) 9 μs; and (**d**)12 μs in 5-AI and 5-plain fabrics.

**Figure 13 materials-14-00479-f013:**
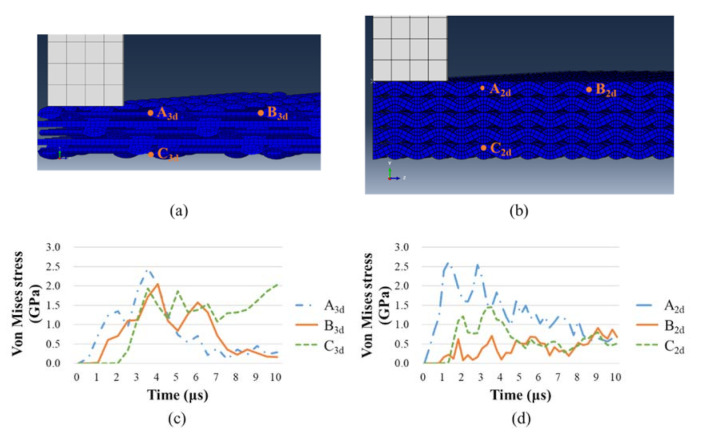
The node positions through the weft direction of (**a**) 5-AI and (**b**) 5-plain; and the stress response of the nodes in (**c**) 5-AI and (**d**) 5-plain.

**Figure 14 materials-14-00479-f014:**
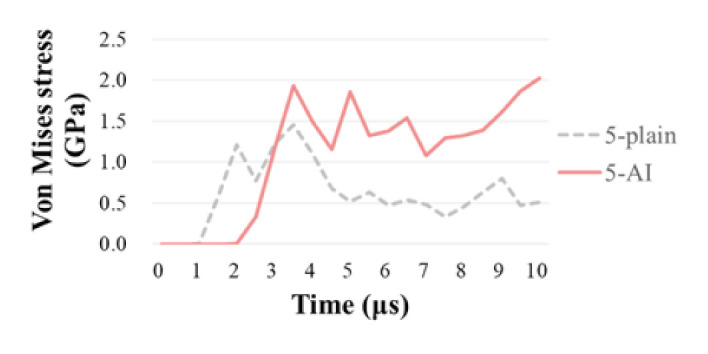
Stress response of node C_3d_ and C_2d_ in 5-AI and 5-plain.

**Table 1 materials-14-00479-t001:** Fabric specifications.

Fabrics Type	Yarn Density (tex)	Spacing of Warp Yarns (mm)	Weft Density (picks/cm)	Thickness (mm)	Size (cm × cm)	Areal Density (g/m^2^)
5-AI	158	0.83	37	1.98	10 × 10	814
3-plain		0.83	12	1.77	5 × 5	1162
5-plain		0.83	12	2.95	5 × 5	1937
4-TTAI		0.83	30	1.67	15 × 15	684

**Table 2 materials-14-00479-t002:** Material properties of FE model.

Materials	Mass Density (kg/m^3^)	Young’s Modulus (GPa)	Poisson’s Ratio	Yield Stress (GPa)	Fracture Strain (%)
Para-aramid yarn	1440	72	0.35	2.9	4.28
steel	7800	209	0.35	-	-

**Table 3 materials-14-00479-t003:** The ballistic performance of fabric models.

Fabric Type	Residual Velocity (m/s)	SEA (J·cm^2^/g)	Peak Accelaration (×10^6^ m/s^2^)
5-AI	410.75	392.97	−4.63
3-plain	428.50	208.91	−4.33
5-plain	383.15	223.97	−8.97

## Data Availability

The data presented in this study are available on request from the corresponding author.

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
