# Peer review of "Numerical Study on the Effect of Z-Warps on the Ballistic Responses of Para-Aramid 3D Angle-Interlock Fabrics"

_materials, 2021, doi:10.3390/ma14030479_

Round 1

Reviewer 1 Report

The paper on the Numerical Study on the Effect of Z-Warps on the Ballistic Responses of Para-aramid Fabrics, provides an interesting study with sound results about the comparative effect of the z-warps on fabrics. The paper is correctly written and organized, and provide the key information to the Readers. 

Still, some points must be clarified by AUTs, as well as a better description of some results. Reviewer believes that the review comments below will be reasonably easy for AUTs to answer. The modified draft would be a clear candidate for publication. 

# AUTs present a paper following a similar analysis scheme as in:  

Yanfei Yang, Xiaogang Chen, Study of energy absorption and failure modes of constituent layers in body armour panels, Composites B 98,
2016 https://doi.org/10.1016/j.compositesb.2016.04.071

also signed by some of the present AUTs. However, it is not cited. This reference can be quite useful for Readers to find information of some details appearing in this paper.  

# AUTs mention some references that use meso-scale models. it would be interesting if AUTs comment, briefly, how the model presented here compares to those meso- alternatives: is more-less complex in details? is more-less realistic? is more-less efficient in terms of computing time?  what cases can be better addressed with the methods ?  

that information will help Readers to better appreciate the analysis model in context with the models.

# the briefly mentioned refs 17 and 18 seem that already provided enough information about the topic discussed in this paper. it is important that AUTs declare what is new in this work, how it complements those references.

REVIEW DETAILS

L35-ff

why these fabrics ? are used in typical ballistic applications as mentioned through the introduction section? otherwise, are the fabrics basic patterns for a general comparison?

L83-ff  

it is confusing: fig-2 has no detailed information on the fabrics. Consider better to connect with Fig-1, types and names.

Figure 2

(L85 ...as shown in figure 2.)

consider to re-locate the figure 2 AFTER figs 3 and 4. it will be much more clear what it is shown after some more descriptions.

caption: mention that b) and c) correspond to 1/4 geometry. also mention that both the fabric and the projectile, in the center of the fabric, are shown.

L88

...only 1/4 model was established... 

can be confusing: consider to re-write as:

...only 1/4 of the fabric and projectile geometry was modeled to save ... 

table-1 and 2

values on the table:

why are these values a reference ? are comparable with the values used in other references ? are comparable with something specific in the market ? otherwise the values appear as arbitrary for the model. 

L89-ff

...The areal density and thickness of 3-plain and 5-AI are close, and are used to compare the energy absorption efficiency of the two structures. 

it can help to connect this dependency with the discussion in the introduction.  maybe following ref 16? 

L91-ff

...The 5-plain and 5-AI have the same number of weft layers, so the analyses based on the number of weft layers will be more intuitive. 

this is hard to see, according to fig-1:  (a) 5 shows weft layers  and (c) shows 10 weft layers...

it may be only the size of the weft in the figures a)-d) in respect to b)-c). but it results quite odd. more input is needed to justify this comparison. similar density? nominal thickness?  

L94-ff and Fig-3

...The geometric parameters of the yarns and the geometric relationships of the fabrics can be obtained based on the measurements and are shown in Figure 3. 

AUTs mention measured values with no source of information. is connected with ref-26? the values on the tables ? specify the source, and connect also with the information as required in section-1: why these fabrics ? are used in applications as mentioned? are basic fibers for comparison ?

FIG-3

the information on weft of AI fabrics is not shown. is it similar to (b) ? 

L106-ff and Fig.4 

Mesh sensitivity studies for various element sizes have suggested that using  ten solid elements across the yarn section is sufficient for this analysis. 

this is no visible in Fig-4. it is not necessary to show the same, but then specify in Figure-4 how many elements are visible, so Readers can get a better view of the problem and model.  

FIG-4

caption: specify the fabric of the model of the picture.

and number of visible solid elements across the yarn section.

L109

...In the model, a global friction coefficient is set to be 0.15. 

reference for this friction value. consider that AUTs have mentioned that yarn-yarn friction is critical, according to refs 13-14.

consider to comment (in the conclusions ?) about the consequences that deviations in this value may induce in the results. the importance of validating this friction values, etc   

L127-ff

on the validation based on energy loss test. 

AUTs should discuss or comment, even briefly, why this simple information can serve as reference for a FE evaluation including a complex fabric model. it is unclear that such simple comparison is really providing a sound comparison as to serve as model check.   

section 3

the intro to section 3.1 is confusing to the general Reader. however, AUTs make a great intro in section 3.2. please, consider moving the 1st line on 3.2 as introductory lines to the section 3:

During the impact process, the fabric will absorb and dissipate the impact kinetic energy carried by the impact projectile in the form of vibration, deformation or breakage, and friction. 

(the sections 3.1, 3.2 ...)

section 3.1

consider to make a simple intro kind of:

The model analysis establish the time start at the projectile input, with nominal velocity, and the results show the velocity change in time through the fabric. The residual velocity is defined...  

L143-ff 

this paragraph is quite confusing. it is really hard to follow the information in the text and the figure 6. please, consider to explain in a more straight style.

L144

...changes less than 0.05 m/s within 3μs 

this is an acceleration: why is not connected with figure 6-b ?

L147

The projectile residual velocity of 3-plain is 4.3% higher than that of 5-AI at a 42.3% larger weight 

4.3% larger... what are the values? why is not mentioned or indicated in the figures?

the next section 3.2 will provide SEA values. however, AUTs did not mention the obtained residual velocities. moreover, fig-6 is only for 2 of the cases in fig-7. AUTs should provide consistent information.

consider to complete fig-6 with the  5-plain results. in any case, mention in the text the values measured as residual velocities for the 3 cases. 

figure-6-A

is Y-axis the residual velocity ?

this is 'the velocity'. the residual velocity is defined by a derivative... it is not this plot.

specify that the result is defined for impact speed 480m/s. 

L172-ff

the paragraph is quite obscure. AUTs mention phenomena that seems to happen without any control by AUTs. eq 3-3 and 3-4 mention 'other' causes of energy loss. what is this? what does the model out of control of the AUTs ?? 

it may be a small value, but AUTs should be capable to manage the behavior of the calculation method applied. even if it is a numerical residue lost in the calculation loops, it should me accounted for as part of the analysis.  it is also helpful to state how much, in %, is this 'others' component. 

indeed, in fig-8 the 'other' component seems to become zero (!?)

L179

...FD is relatively small and at an equivalent amount among these three fabrics. 

provide a range : 15 to 20% of E_tot?

section 3-3 & fig-10 and fig-12

AUTs compare results of 5-AI and 5-plain. 3-plain is missing.

same in figures 10 and 12.

at least mention (as done in fig-9) if 3-plain is is similar or not. otherwise, why only comparing 5-AI and 5-plain ?

L212

...The study of the impact stress distribution of the two fabrics will prove this speculation. 

if AUTs make this study, add, 'as it will be shown in section 3.5 (?!)'.

L216-222 

AUTs present deformation, but refer again and again to stress. Figures 11-12-13 and section refers to 'deformation'. keep the concepts to avoid too much distraction. consider to keep DEFORMATION in the sentences. 

if it was necessary, connect concepts explicitly:

... the deformation (and therefore, the stress)... 

fig-11 caption

mention that only 1/4 of the geometry is shown, and that impact is located at the upper-left corner. it may be obvious, but Readers will appreciate the help to focus into the relevant information.  

L217

A quarter of the fabric panels were examined in both warp and weft directions, as shown in Figure 11. 

in figure: what is shown ? warp or weft deformation? does it simply depends on the axis?  

why IN-PLANE ? please comment briefly what deformation is evaluated in fig-11.

L220

... 5-AI has a wider range of stress spread in weft direction...

how is it possible to see this in figure 11 ? please, specify this details (as weft direction) in the caption... otherwise it is not possible to follow the discussion.

L223-ff and fig-12

3-plain results are missing. why?

it would reinforce statements as (L225)

...the deformation of 5-plain are equivalent along both warp and weft directions... 

L231-ff and fig-13

it is difficult to connect the information. AUTs mentioned before in-plane deformation (to be better defined...), and now 'side-view of deformation along weft direction'. is this out-of-plane (along weft direction)?

fig-13: the red stuff:  it is the projectile (!)?  mention that in the caption to provide a more effective information and to help Readers to immediately focus into the key information. 

L251

...The longitudinal wave propagates along the fiber axis, 

what is the fiber axis in a 3D fabric ? 

L260

... The yarns at the impact face failed early, ...

complete the information with the previous results: 

... The yarns at the impact face failed early (compare with Figure 10)...

L260-ff and fig-14

(as before)  3-plain results are missing. why?

it would reinforce the statements about 5-plain local narrow stress distribution.

L274

... According to the analysis above, the in-plane stress distribution in the 3D fabric is uneven, and the weft direction has a larger deformation range and more resistance time than the warp direction. 

this may be obvious to the AUTs, but AUTs must explain why they make this statement, according to the 'analysis above'.

L276

AUTs used undefined information. please state promptly the information used. consider to modify the sentence:

For further comparison, 3 nodes were selected as in Figure 15:  A3d, B3d and C3d from 5-AI fabric along the weft direction, and A2d, B2d and C2d from 5-plain fabric. It can be seen...

L299

...The result provides guidance for the following work.

which work is refereed ? a future publication? 

L299-ff

It is possible to appropriately increase the interlacements between the weft layers and the yarn-yarn friction by adding straight wadding warp yarns ...

it seems that AUTs refer to a new fabric: does it exists ? is it possible to make it ? 

Otherwise, refer to a specific fabric knitting in the market as a better option to increase impact performance. however, note that only a new calculation would probe the hypothesis. this cannot be requested to AUTs in this study. therefore, it would only be a guess...

DETAILS...

L143 ...when IT changes...

L184 ... the primary yarns that ARE in direct 

L225 ...the deformation of 5-plain IS equivalent along both warp and weft directions... 

Reviewer 2 Report

Dear Author 

  1. The author must check the format prescribed by the journal. 
  2. There is no much information for why you did this work, I mean objective of this work is missing. 
  3. both 5.5 mm and its mass is 1.0 g// how?
  4. fabric at the impact velocity of 480 m/s/ why specific velocity has been chosen?
  5. Why no error bars used in Fig 5. 
  6. Compared with 3-plain, the SE of 5-AI is higher, which means that yarns of 5-AI can store more energy in a manner of elastic deformation, and convert it into other forms of energy (such as KE and FD) for a gradual release after reaching the maximum value// How ? 
  7. It is speculated that Z-warps in 5-AI provide structural support for the panel through-the-thickness and bind all of the weft layers together,// How?

Round 2

Reviewer 1 Report

The draft includes a tough revision of the different points as suggested in the Revision. The new information, figures and added discussions improve the overall description of the case. 

the Reviewer insists on a couple of details, listed below, for the consideration of the Authors, but this is just a minor revision. Otherwise the paper is ready for publication.

L114

please, consider to better point out the information source:   

The material properties are shown in Table 2, according to the results in [22]. 

L150-ff

Impact and residual velocities validation. It remains unclear that this comparison would be useful to set a sharp check for the model itself. 

I suggest to add a new set of data, corresponding to one of the other fabrics. This will show if a different fabric causes a clear difference in the trend or values. Therefore, it would be reinforced the message of the sensitivity of this comparison of model basic data, as well as model differences in fabrics. 

If this comparison is not neat, then do not include it. It will be up to the Readers to consider the limits of this validation.   

L206-ff

Certainly Authors did mention in the original version the required information about 'missing' energy in the balance equations. however, to make it easier for Readers to have the full picture while reading section 3.2, please repeat the message at this point, may as:

(L208) KE.. IE... FD ... and others. As mentioned before, the other sources of energy dissipation (projectile deformation, fiber intermolecular friction, air resistance and acoustic losses) were assumed to be negligible. 
